Elevational surveys of Sulawesi herpetofauna 2: Mount Katopasa on the Eastern Peninsula of Sulawesi island, Indonesia

http://orcid.org/0000-0003-1167-5233 Krone Isaac W. 1 2 ikrone@berkeley.edu
Karin Benjamin R. 1 3
http://orcid.org/0000-0001-5014-2507 Frederick Jeffrey H. 1 4
http://orcid.org/0009-0004-9904-5168 Amini Sina S. 1 2
http://orcid.org/0000-0003-0976-9337 Scarpetta Simon G. 5
Hamidy Amir 6 7
http://orcid.org/0000-0003-4004-7906 Anita Syahfitri 7
http://orcid.org/0000-0002-6887-9352 Riyanto Awal 7 8
Arida Evy 6
Laksono Wahyu Tri 6 7
http://orcid.org/0000-0003-3638-6016 Arifin Umilaela 1 9
http://orcid.org/0000-0003-1420-6344 Bach Bryan H. 10
http://orcid.org/0009-0009-1212-176X Bos Collin 1
Jennings Charlotte K. 1
http://orcid.org/0000-0003-1524-7287 Stubbs Alexander L. 1
Peterson Kinsington 1
Shi Sucheng 1
http://orcid.org/0000-0002-9562-5585 McGuire Jimmy A. 1 2
1 Museum of Vertebrate Zoology, University of California, Berkeley , Berkeley, CA , United States
2 Department of Integrative Biology, University of California, Berkeley , Berkeley, CA , United States
3 Department of Environmental Science, Policy, and Management, University of California, Berkeley , Berkeley, CA , United States
4 Department of Biology, University of Kentucky , Lexington, KY , United States
5 Department of Environmental Science, University of San Francisco , San Francisco, California , United States
6 Research Center for Applied Zoology, Badan Riset dan Inovasi Nasional , Cibinong, West Java , Indonesia
7 Laboratory of Herpetology, Museum Zoologicum Bogoriense, Badan Riset dan Inovasi Nasional , Cibinong, West Java , Indonesia
8 Faculty of Mathematics and Natural Sciences, University of Indonesia , Depok, West Java , Indonesia
9 Center for Taxonomy and Morphology, Leibniz Institute for the Analysis of Biodiversity Change , Hamburg, Hamburg , Germany
10 California Institute for Quantitative Biosciences, University of California, Berkeley , Berkeley, California , United States
Manjarrez Javier
Electronic publication date: 2025 Sep 25
Publication date: 2025
Volume: 13
Electronic Location ID: e20024
Received 2025 Mar 27; Accepted 2025 Aug 11
Copyright: © 2025 Krone et al.
Copyright year: 2025
Copyright holder: Krone et al.
License: This is an open access article distributed under the terms of the Creative Commons Attribution License, which permits unrestricted use, distribution, reproduction and adaptation in any medium and for any purpose provided that it is properly attributed. For attribution, the original author(s), title, publication source (PeerJ) and either DOI or URL of the article must be cited.
License URL: https://creativecommons.org/licenses/by/4.0/

Keywords: Southeast Asia, Asia Tenggara, Elevational gradient, Gradien Ketinggian, Herpetofauna, Biodiversity survey, Survey Keragaman Hayati, Walacea, Herpetology, Herpetologi

Funding: National Science Foundation of the United States 1457845 & 1652988 This research was supported by the National Science Foundation of the United States (Nos. 1457845 & 1652988). The funders had no role in study design, data collection and analysis, decision to publish, or preparation of the manuscript.

==============================
The unique geologic and biogeographic history of the island of Sulawesi in Indonesia has produced one of the world’s most endemic biota, much of which is still unknown to science. The biogeography of the island is influenced both by its unique shape, a confluence of four peninsulas, as well as its extremely mountainous topography, and its upland ecosystems are poorly understood by science. Here, we report the second full-mountain checklist from a series of herpetological surveys of Sulawesi’s Mountains conducted by an international team of researchers. During 3 weeks of work on Gunung Katopasa, a 2,825 m asl peak on Sulawesi’s Eastern Peninsula in the province of Sulawesi Tengah, we recovered 45 species of reptiles and amphibians. Among these, we believe that at least eight represent undescribed species. Our efforts to survey Gunung Katopasa highlight that greater herpetological diversity will likely be discovered upon future assessments of species richness and abundance on the Eastern Peninsula of Sulawesi.

Introduction

The island of Sulawesi harbors a unique and endemic biota that evolved across a dynamic paleo-archipelago, recently accreted into a single island that sits within a major biogeographic transition zone (Koch, 2012; Nugraha & Hall, 2018; McGuire et al., 2023). Positioned between the Sunda shelf, the furthest extent of the Asian continental shelf, including Java, Bali, and Borneo, and the Sahul shelf, the continental shelf surrounding Australia and New Guinea, Sulawesi and its neighboring island chains form a biogeographic intergrade zone, known as Wallacea, in which Sahul and Sunda biota mix. Deep ocean channels surrounding Wallacea represent major biogeographic barriers to the colonization of these islands, which have never been connected to continents by land. To the west, Wallace’s line follows the deep trench between Sulawesi and Borneo, continuing south to separate Bali from Lombok, and to the East, Lydekker’s line follows a trench that separates the Lesser Sundas, Tanimbar Islands, and Maluku Islands from the Sahul shelf.

Sulawesi has a complex and dynamic paleogeographic history. Emergent land is thought to have first appeared when West Sulawesi rifted from the Sunda Shelf margin ~45 Ma, resulting in the formation of the Makassar Strait, a known biogeographic barrier to dispersal between the Sunda Shelf and more easterly landmasses (Wilson & Moss, 1999) that is a primary contributor to Wallace’s Line (Ali & Heaney, 2021). While the presence of continuous land between 45 and 23 Ma remains uncertain, with submersion possible during this interval, land has certainly been present since the Sula Spur collision ~23 Ma. Recent paleotectonic reconstructive analyses by Nugraha & Hall (2018) indicate that Sulawesi existed as one or two paleo-islands at 23 Ma. Subsequently, several islands formed over the next 20 million years ultimately resulting in as many as 13 paleo-islands by 2 Ma; these smaller islands were then further uplifted, enlarged, and ultimately amalgamated, resulting in the present configuration of the island. This tectonic history of paleo-island accretion is considered to have played a critical role in facilitating the formation of at least seven primary areas of endemism (AOE) on Sulawesi (Evans et al., 2003; Setiadi et al., 2011; McGuire et al., 2023).

Recent wildlife surveys of Sulawesi have revealed large numbers of undescribed species and extensive lineage diversity, highlighting the need for thorough sampling across all of Sulawesi’s areas of endemism (Evans et al., 2003; Setiadi et al., 2011; Handika et al., 2021; Esselstyn et al., 2021; McGuire et al., 2023; Karin et al., 2023; Frederick et al., 2023). In this study, we report the results of a multi-week herpetological survey conducted in 2018 on Gunung Katopasa, a 2,825 m mountain on Sulawesi’s Eastern Peninsula. This expedition was part of a multi-year project aiming to survey vertebrate diversity on mountains across Sulawesi and is our second such report (see Karin et al., 2023 regarding our survey of Gunung Galang).

Gunung Katopasa sits near the border of the area of endemism associated with Sulawesi’s Eastern Peninsula. The Eastern Peninsula is currently connected to the rest of Sulawesi by relatively low-lying habitat (generally under 800 m above sea level (asl)) that corresponds to a biogeographic break between clades of macaque monkeys, Celebes toads, and Draco flying lizards (Evans et al., 2003; McGuire et al., 2023). Higher-elevation habitats on the Eastern Peninsula are thought to be geologically young. Though some rocks of the Eastern Peninsula have been subaerial since at least 20 Ma, Pliocene geologic deposits in both the southern and northern portions of the Eastern Peninsula suggest that uplift resulting in modern high-elevation environments initiated ~3 Ma, coinciding with the fusion of Sulawesi’s Eastern and Southeastern Peninsulas with the Island’s Central Core region (Nugraha & Hall, 2018; Nugraha, Hall & BouDagher-Fadel, 2022). Elevations above >2,000 m are not inferred until closer to 2 Ma, and reworked rocks from the north and Quaternary reefs several hundred meters above sea level in the south record further uplift during the Pleistocene c. 1 Ma at least until 300 Ka (Nugraha & Hall, 2018; Nugraha, Hall & BouDagher-Fadel, 2022).

Thus, mountains of the Eastern Peninsula formed over the past 3 Ma, have been gaining elevation until very recently, and certainly did not attain close to their highest elevations until starting around ~2 Ma. We therefore expect high-elevation taxa on Gunung Katopasa to have diverged from low-elevation counterparts within the last few million years, or to represent the result of range-expansions and/or competitive interactions that pushed lowland lineages into higher-elevation habitats. Glacial cycles during the Pleistocene and associated effects on sea level (falls of up to 150 m and rises of up to 50 m) may also have affected elevational biodiversity gradients over the past 1 Ma (Nugraha & Hall, 2018). The history of paleo-island formation and aggregation and recent uplift suggests that the mountainous areas of the Eastern Peninsula should harbor many endemic lineages and unique ecological communities.

At 2,825 m in elevation, Gunung Katopasa is the ninth tallest mountain on Sulawesi and the second tallest on the Eastern Peninsula (only the adjacent Gunung Kandela, at 2,870 m is taller). Gunung Katopasa is located in the north-western portion of the peninsula ca. 20 km from the Gulf of Tomini and is connected to the larger Gunung Kandela to the south by a ca. 2,000 m elevation ridge, suggesting that high elevation endemics could be shared between these two mountains. The Katopasa and Kandela massifs are separated from other prominent mountains on the Eastern Peninsula (e.g., Gunung Tambusisi (2,422 m) and Gunung Tokala (2,593 m)) by an upland plateau dissected by many small rivers and drainages. These and other river systems such as the Bongka River may represent additional biogeographic barriers on the Eastern Peninsula (see Froehlich & Supriatna, 1996; Evans et al., 2003) that could influence population structure or local patterns of endemism.

To our knowledge, there has been limited effort to survey herpetofauna at higher elevations on the Eastern Peninsula before our expedition. Previous efforts led by Rafe Brown in 1998 and Djoko Iskandar in 2009 surveyed Gunung Tompotika, a 1,540 m mountain at the Eastern end of the Eastern Peninsula. These expeditions collected numerous specimens and resulted in the discovery of several new species Gekko iskandari (Brown, Supriatna & Ota, 2000), Cyrtodactylus batik (Iskandar, Rachmansah & Arifin, 2011), and Occidozyga tompotika (Iskandar, Arifin & Rachmansah, 2011), but the mountain’s size would have prevented these teams from sampling much area above 1,400 m, leaving the high-elevation herpetofauna of the eastern Peninsula quite poorly known. Therefore, our sample of the Gunung Katopasa fauna is almost certainly the most extensive high-elevation herpetological collection from the Eastern Peninsula.

Materials and Methods

Field surveys

This expedition was part of a large trans-national collaborative project between several institutions including the National Research and Innovation Agency (BRIN, formerly LIPI, including many researchers from Museum Zoologicum Bogoriense (MZB)), Indonesia; Tadulako University, Indonesia; Museum of Vertebrate Zoology (MVZ), USA; American Museum of Natural History (AMNH), USA; Museum Victoria (MV), Australia; and Louisiana State University (LSU), USA. Our team consisted of six herpetologists, but there were 26 scientists in total and several local guides conducting fieldwork on this expedition. All work conducted as part of this research was approved by the Institutional Animal Care and Use Committee of the University of California Berkeley (AUP-2014-12-6954-2) under JAM. Research and collection permits, and materials sharing agreements were administered by LIPI and RISTEK-DIKTI (now BRIN) and granted to JAM (RISTEK-DIKTI permit 213/SIP/FRP/E5/Dit.KI/VIII/2017).

Our expedition to Gunung Katopasa took place between 6 August and 6 September 2017, with field surveys and collections between August 14th and September 4th. On 12 August, we traveled to Palu in the early morning where we purchased provisions for our field team. On 13 August, we drove to Desa (village) Mire on the Eastern Peninsula, which was the jump-off point for our Gunung Katopasa fieldwork. On 14 August, we scouted the low-elevation camp and set that camp at 375 m elevation, a short hike up the main river from Desa Mire. We later set a lower mid-elevation camp at 750 m elevation and a higher mid-elevation camp at 1,366 m elevation. We were able to survey between ~250 m elevation and ~2,575 m elevation using these camps. Access to these elevations was facilitated by the presence of a trail used by local people for accessing marketable forest products such as dammar, but also used by climbing enthusiasts to summit Gunung Katopasa. Sampling strategies involved hand collection at all elevations, three drift-fence/pitfall bucket arrays at 434 m (13 buckets), 1,138 m (15 buckets), and 1,453 m (15 buckets), and five glue trap lines on select ground substrates and tree buttresses from 382–411 m (30 traps), 1,380–1,432 m (35 traps), 1,490–1,511 m (30 traps), 1,800–1,900 m (30 traps), 2,015–2,170 m (30 traps). We conducted targeted searches for herpetofauna during the day and at night (more extensively during the latter), and incidentally encountered herpetofauna during hikes between camps and regular forays to our pitfall and glue trap lines during the day and night. These techniques have been demonstrated to be effective in fully surveying Sulawesi’s herpetofauna (Gillespie et al., 2005). Although we did not work out of the temporary 2,575 m high camp, we did a collecting foray to the ~2,575 m Pos 9 first summit, but collected no herpetofauna above 1,650 m (although JAM heard Oreophryne sp. as high as 2,100 m and non-herpetologist members of our team may have heard them at the 2,575 m lower summit). We worked out of the low camp from 14 until 20 August and out of the upper mid-elevation camps from 21 August until 4 September, with a few team members briefly working out of the lower-mid elevation camp as well.

We obtained liver tissue samples in RNA later for all specimens collected, which were frozen in liquid nitrogen within 48 h. We swabbed all frogs for Batrachochytrium dendrobatidis fungus and took blood smears to screen for Plasmodium and other haemosporidian blood parasites for most specimens. Gut contents for Limnonectes fanged frogs were extracted and preserved in 95% ethanol.

All specimens collected are housed either at the MZB (half of the specimens, plus most singletons) or at the MVZ. Tissue samples for each specimen were divided with a subsample provided to each collection. An appendix containing elevation, location, time, and environmental data for our specimens is available.

Statistical analyses

We carried out all analyses in R 4.2.3 (R core group). Our code largely follows that of Karin et al. (2023). Our analyses use the vegan package (Oksanen et al., 2022) to calculate chao and Jackknife diversity estimates, the divDyn package (Kocsis et al., 2019) to calculate range-through diversity, and the leaflet package to map the specimens, using terrain tiles from ESRI. All of our analyses are based on data from the field catalog of specimens.

Our data categorization follows Karin et al. (2023). We broadly categorized herpetofauna as frogs, snakes, or lizards. We classified elevations below 700 m as “lowland,” elevations from 700 to 1,400 m as “middle elevation,” and elevations above 1,400 m as “high elevation.” All specimens with location data had associated elevation data; however, we checked the accuracy of the reported elevations against a digital elevation model (DEM) (AWS Open Data Terrain tiles at a 9.6 m resolution) of the region using the elevatr package (Hollister et al., 2022; AWS, 2025), because we have found elevation readings on GPS units to be less precise than latitude/longitude readings in topographically complex field environments. Though we identified 14 instances in which the DEM and reported elevations differed by more than 50 m, all associated specimens were found below 700 m elevation, meaning that these potential inaccuracies in our field data would not affect the observed pattern of large-scale elevational turnover. Despite these outliers, our reported elevations overwhelmingly correlate with the elevations of the digital elevation model (R2 = 0.997, p < 0.0001). Interestingly, discrepancies between reported and model elevations do not correlate with our reported coordinate error (R2 = 0.007, p = 0.26) or the corresponding range of elevations from the DEM within the error radius (R2 = 0.002, p = 0.54).

To assess the health of the Gunung Katopasa ecosystem and to better understand trends in habitat integrity, we produced a “Dashboard” for the location via Global Forest Watch data browser (World Resources Institute, 2025a). We based this dashboard on a rough polygonal outline of Gunung Katopasa and its watershed based on topographic maps (ESRI world topo map). We also produced a “Dashboard” for the Tojo Una-Una regency (World Resources Institute, 2025b). These dashboards provide an analysis of forest composition and change since the year 2000, which we used to investigate threats to and changes in the ecosystem of Gunung Katopasa.

We produced a map of our survey and the surrounding area using the terra package (Hijmans, 2025), incorporating elevation data from AWS digital elevation models (AWS, 2025), forest cover data from Hansen et al. (2013) and road and river layers from OpenStreetMaps, accessed via the osmdata package (Padgham et al., 2017).

Results

Habitat

The immediate vicinity of Desa (Village) Mire is composed mostly of highly disturbed agricultural land, some of which is planted with corn while other areas consist of coconut palm plantation. Below 250 m elevation there is virtually no natural habitat. Between 250 and 700 m, the habitat is mostly disturbed but there are patches of secondary gallery forest through which clear, high-flow rocky streams pass, and we made collections in these habitat patches. The trail to the summit of Gunung Katopasa bypasses these lower elevation forest patches, instead passing through corn fields, grassland, coconut plantation, and forest scrub before entering a section of more developed secondary forest at about 750 m elevation that appeared to have recently burned when we performed a scout trip in March 2016 and which still showed evidence of that event during our expedition (see Fig. 1). At ~1,150 m, the forest improved markedly; at this elevation, the forest appears to have been selectively logged, retaining many large trees and complete overhead canopy. The habitat transitions to primary (or at least mature, selectively logged) lower montane forest with little sign of human disturbance at about 1,400 m elevation where we established our high camp.

Figure 1 Topographic shaded relief map of survey area; inset shows location on Sulawesi.

Specimen localities are marked according to the taxon of the specimen. Green shading indicates the extent of canopy cover as of the year 2020; areas in dark gray indicate forest cover loss in 2015–2016, much of which was due to fire. Black contour lines designate the 700 m intervals used to define the elevational bands.

As is often the case for Sulawesi mountains, there were few streams for us to sample that were accessible from the primary summit trail. A small trickle stream immediately below our camp was accessible. At 1,470 m elevation (Lat = −1.18932, Long = 121.44165), there was a remarkable sedge swamp wetland unlike anything we have observed on other mountains (see photo: Fig. 2F) that was inhabited by a new species of Rhacophorus, since described as R. boeadi (Hamidy et al., 2025), as well as by the rhacophorid frog Polypedates iskandari, and the snake Ptyas dipsas. The sedge swamp was occupied by a huge breeding aggregation of R. boeadi when we arrived at the site on August 21st, with hundreds of individuals actively engaged in nighttime breeding, and extensive numbers of foam nests in the vegetation and on a large log in the middle of the swamp. This aggregation with many calling males and amplectant pairs was completely absent 2 days later, on August 23rd.

Figure 2 In situ expedition photographs.

(A) Gunung Katopasa from the village of Mire. (B) Lowland forest consisted of mixed agricultural land. (C) Fast-moving lowland stream cut deeply into the bedrock. (D) View from the mid camp overlooking the burned area. (E) View of high elevation forest with stunted, moss-covered trees. (F) The sedge swamp which hosted a large breeding population of the newlydescribed Rhacophorus boeadii. (G) Higher elevation mossy forest above 1,400 m. Photos by B.R.K.

At Pos 6 (1,670 m), a small 1 m wide rocky seep with limited flow was inhabited by Rhacophorus edentulus and Occidozyga semipalmata (note that our Occidozyga could be O. tompotika, but we refrain from referring our samples to that species because our unpublished data indicate that O. semipalmata is a species complex on Sulawesi often including unique low and high elevation lineages on the same mountain, and our samples are from substantially higher elevation than the type locality of O. tompotika). From there, a steep waterfall dropped into a slot canyon containing a small stream with light flow and occasional deeper pools. This stream course included a series of difficult to traverse waterfalls and cliffs that descended steeply toward a larger stream that we could hear in the distance but not see. We found only O. semipalmata in this stream. Above Pos 6, there were no streams accessible from the trail, and much of the trail through lush mossy forest was extremely steep, requiring extensive hand-over-hand climbing using exposed roots as hand-holds. At ~1,860 m, there was a small plateau, but after this plateau it was mostly nearly vertical again until Pos 7 (2,015 m), above which was another small plateau. The absence of flowing water above 1,670 m explains why the only species we observed above this elevation (by voice) was Oreophryne sp., a direct developer that does not require free water for reproduction.

Herpetofauna

In total, we collected 262 specimens representing 45 species of amphibians and reptiles over 21 days and several hundred worker-hours of survey effort, including 14 species of frogs, 15 species of lizards, 15 species of snakes (Table 1), and one species of turtle, Cuora ambionensis. During our expedition, we also observed but did not collect the frog Papurana celebensis and the lizards Lamprolepis smaragdina and Varanus salvator in the lowlands, and the snake Ptyas dipsas at the sedge swamp. Among the 23 Sulawesi endemic species we observed, we suspect that two (undescribed Oreophryne sp. and Calamaria sp.) may be endemic to the Eastern Peninsula, but some herpetofaunal species are endemic to both the Eastern and Southeastern Peninsulas (e.g., Oligodon tolaki (Amarasinghe et al., 2021)) and these may be as well. The majority of our specimens were collected in the lowlands, with an especially large proportion collected out of the 375 m camp. Though the habitat gap we found between 800 and 1,350 m prevented us from thoroughly surveying the middle elevation of the mountain, higher elevation habitats proved more intact, though with a depauperate herpetofauna. The high-elevation sample included four species of squamates (Rhabdophis chrysargoides, Calamaria sp., Eutropis macrophthalma, and Sphenomorphus zimmeri), and five species of frogs (Polypedates iskandari, Rhacophorus boeadi, R. edentulus, Occidozyga semipalmata, and Oreophryne sp.) (Fig. 3). We collected specimens as high as 1,660 m (Table 1), though we heard Oreophryne frogs calling at 2,100 m and at the 2,575 m puncak (Fig. 1—“first summit”) and suspect that they are present all the way to the summit of Gunung Katopasa at 2,825 m based on our experiences with Oreophryne on other mountains.

Table 1 All frog, lizard, and snake species collected during the expedition with corresponding numbers of specimens, ranges of snout-vent length (SVL; mm), weight (gram), and elevation (m a.s.l.).

Group	Species	n	SVL	Mass	Elevation	
Frog	Chalcorana mocquardi*	1	45	4.48	365	
Frog	Duttaphrynus melanostictus	3	78–95	37.15–57.29	262–765	
Frog	Ingerophrynus biporcatus	1	31	2.13	296	
Frog	Ingerophrynus celebensis*	2	16–80	0.33–29.86	276–409	
Frog	Kaloula baleata	3	49–65	9.96–27.47	297–434	
Frog	Limnonectes sp. “1”*	15	25–42	1.6–6.82	377–738	
Frog	Limnonectes sp. “G2”*	12	45–71	8.39–26.26	280–670	
Frog	Limnonectes sp. “I”*	18	14–170	0.26–349.8	261–1,339	
Frog	Limnonectes sp. “T Yellow”*	5	40–56	5.66–14.59	1,335–1,364	
Frog	Occidozyga semipalmata*	11	19–37	0.82–5.53	1,339–1,639	
Frog	Oreophryne sp.**	25	14–22	0.31–1.02	1,375–1,588	
Frog	Polypedates iskandari*	7	23–65	0.74–11.48	347–1,502	
Frog	Rhacophorus boeadi*	25	21–61	0.67–8.12	1,471–1,521	
Frog	Rhacophorus edentulus*	17	32–40	1.56–3.38	1,276–1,650	
Lizard	Cyrtodactylus jellesmae*	6	53–67	2.16–4.92	318–757	
Lizard	Cyrtodactylus sp.*	4	46–61	1.6–3.5	282–482	
Lizard	Dibamus sp.*	1	142	1.62	301	
Lizard	Draco beccarii*	3	64–68	3.46–3.73	365–441	
Lizard	Emoia caeruleocauda	2	56	3.49–3.55	352	
Lizard	Eutropis macrophthalma*	5	65–128	6.8–57.99	721–1,511	
Lizard	Eutropis multifasciata	4	61–97	5.12–22.06	339–757	
Lizard	Eutropis rudis	18	33–88	1.08–20.04	283–736	
Lizard	Gehyra mutilata	2	70–75	8.76–10.19	694	
Lizard	Hemidactylus frenatus	1	44	2.05	283	
Lizard	Lipinia infralineolata*	2	41–48	1.24–1.37	331–381	
Lizard	Sphenomorphus tropidonotus*	10	35–81.5	1.09–17.65	364–420	
Lizard	Sphenomorphus variegatus*	8	29–53	0.62–3.19	346–461	
Lizard	Sphenomorphus zimmeri*	21	31–78	0.66–10.46	1,366–1,581	
Lizard	Tytthoscincus sp.*	7	31–53	0.62–0.95	352–1,100	
Snake	Ahaetulla prasina	3	685–790	23.84–25.98	293–1,366	
Snake	Boiga irregularis	1	1,566	400	384	
Snake	Calamaria brongersmai*	2	173–192	2.65–3.04	273–670	
Snake	Calamaria sp.**	1	185	2.86	1,522	
Snake	Chrysopelea paradisi	2	515–553	30.94–31.85	210–402	
Snake	Cylindrophis melanotus	1	325	17.74	756	
Snake	Dendrelaphis marenae	1	293	3.3	1,356	
Snake	Hypsiscopus plumbea	1	336	31.42	381	
Snake	Oligodon tolaki*	1	273	9.36	381	
Snake	Ophiophagus hannah	1	–	–	409	
Snake	Psammodynastes pulverulentus	3	296–428	9.3–29.09	305–358	
Snake	Rhabdophis chrysargoides	1	619	87.48	1,490	
Snake	Tropidolaemus subannulatus	1	444	94.22	418	
Snake	Xenochrophis trianguligerus	1	763	219.89	526	
Snake	Xenopeltis unicolor	1	689	260.33	440	
Note:

Species marked with an asterisk (*) are endemic to Sulawesi; species marked with two asterisks (**) are possibly endemic to the Eastern Peninsula. The sole Ophiophagus hannah specimen was an incomplete skin shed, so SVL and mass are not reported.

Figure 3 All frog (green; upright triangles), lizard (light blue; circles), and snake (dark blue; inverted triangles) species and specimens observed on Gunung Katopasa, arranged by maximum elevation.

The upper line plot tallies species richness in 10 m intervals under a range-through assumption. Vertical dashed lines at 700 and 1,400 m demarcate the upper boundaries of the “low” and “middle” elevational bands. The green, long-dashed line represents frogs; the light blue, short-dashed line represents lizards; the dark blue, dotted line represents snakes. Total richness is represented by the solid gray line. Four species observed but not collected are figured here; Papurana celebensis, Lamprolepis smaragdina, Varanus Salvator, and Ptyas dipsas.

Our pitfall traps recovered very few reptiles and amphibians. The pitfall line at 434 m collected five specimens; two Tytthoscincus sp, as well as Sphenomorphus variegatus, Kaloula baleata, and Cyrtodactylus jellesmae. Pitfall traps at 1,435 m recovered two Sphenomorphus zimmeri, as well as one Calamaria sp. and one Oreophryne sp. Compared to our efforts on other Sulawesi mountains, only a moderate number of specimens were caught via glue traps, and the method was even less effective at higher elevations: 15 specimens were collected on traps from 382 to 411 m, 10 specimens between 1,380 and 1,432 m, four specimens on the 30 traps placed between 1,490 and 1,511 m, and no specimens at highest-altitude glue traps placed between 1,800 and 2,170 m. Detailed information on the precise locations and microhabitats in which our specimens were found are available in Table S1.

Sampling efficacy

Both Chao and Jackknife estimators suggest that we observed species composing two-thirds of the total herpetofaunal species on Gunung Katopasa during our 3 week survey (Table 2). According to these estimators, two to four frog species, four to six lizard species, and potentially dozens of snake species were not observed during our surveys.

Table 2 Observed and estimated squamate and amphibian species diversity within taxa and elevational bands.

Excludes four species observed but not collected.

Subset	Specimens found	Observed species	Number of days sampled	Chao estimate	Chao standard error	Second-order Jackknife estimate	Estimated missing species (Chao/Jackknife)	
Total mountain	264	44	21	69.71	16.82	72.27	26/28	
<700 m	131	33	11	51.77	12.33	57.21	19/24	
700–1,400 m	42	16	17	39.53	22.84	32.58	24/17	
>1,400 m	86	9	12	13.12	6.62	13.49	4/4	
Frogs	146	14	18	16.12	3.23	17.83	2/4	
Lizards	96	15	17	18.76	5	20.64	4/6	
Snakes	22	15	11	42.5	26.09	32.53	28/18	

The majority of species that we collected were obtained below 700 m, but we are confident that our collecting did not encompass the full suite of herpetological diversity in this species-rich elevational band, as indicated by statistical estimators (Table 2) and our species accumulation curve (Fig. 4). During prior fieldwork in the lowlands of the Eastern Peninsula, members of our team have collected the frog Fejervarya cancrivora, the lizards Bronchocela celebensis, Gekko smithii, and Emoia atrocostata, and the snakes Indotyphlops braminus and Malayopython reticulatus. The lizards, Gekko iskandari (known from one specimen) and Cyrtodactylus batik are known only from Gunung Tompotika near the terminus of the Eastern Peninsula and could also be members of the Gunung Katopasa fauna along with the house gecko, Hemidactylus platyurus.

Figure 4 Accumulation of specimens over time.

(A) Accumulation of frog, lizard, and snake species over the length of the survey period, plus the cumulative species count; colors and line types follow Fig. 3. (B) Accumulation of all frog, lizard, and snake species found below 700 m (solid line), from 700 to 1,400 m (long dashes), and above 1,400 m (dot-dashes) elevation over the number of days during which specimens were collected in that band.

Forest loss

The majority of land on Katopasa and in its watershed is forested, but forest density has decreased drastically since the year 2000. As of 2023, tree cover has decreased by 114 km2 (17%); half of the tree cover lost has been primary forest. The large-scale burning event that we saw evidence of took place in 2015–2016, removing approximately 10 km2 of forest surrounding the trail to the summit. In these same years, Gunung Katopasa and its watersheds lost more than 40 km2 of tree cover, more than half due to fire (World Resources Institute, 2025a). Large forest fires, most set to clear land for industrial agriculture, burned vast areas of land in Indonesia in 2015 (Purnomo et al., 2019; Edwards et al., 2020) and 2016, and directly burned more than 730 km2 of forest in Sulawesi. During 2015 and 2016, Sulawesi lost more than 3,800 km2 of forest, approximately 2.8% of its forest area circa the year 2000 (World Resources Institute, 2025c). Deforestation appears to have been particularly severe in Tojo Una-Una regency in these 2 years, in which areas of forest equivalent to 10% of the forest cover circa the year 2000 were lost, with fire being a major contributor (World Resources Institute, 2025b).

Since our expedition in 2017, the Katopasa watershed has lost more than 22 km2 of additional forest, equivalent to 3.3% of its year-2000 forest cover. Forest loss has occurred mostly in small patches at low elevations, a pattern consistent with clearing for agriculture. Afforestation in the area has been negligible throughout the 21st century; the 3.4 km2 of tree cover gained is associated with a few lowland patches and regrowth at the edges of a large landslide area on the West slope of the mountain (World Resources Institute, 2025a).

Discussion

Gunung Katopasa harbors a rich herpetofauna, likely comprising more than sixty species (Table 2). Hundreds of specimens were collected by our survey, likely representing more than two-thirds of the total herpetofaunal species richness on the mountain. Our recovery of more than half a dozen undescribed species on Gunung Katopasa emphasizes the need for more scientific work on the fauna of Sulawesi Tengah.

We are confident that there is elevational stratification in Gunung Katopasa’s herpetofauna despite the lack of forest on our transect between 800 and 1,400 m. Middle-elevation specialist species are known from other mountains (Karin et al., 2023), but our survey did not recover any species definitely unique to the middle elevation band between 700 and 1,400 m (Cylindrophis melanotus is known to range from 0–1,200 m elevation (de Lang & Vogel, 2005)). Four species of frogs, Rhacophorus boeadi, Oreophryne sp., Occidozyga semipalmata, and Rhacophorus edentulus and the skink Sphenomorphus zimmeri all appeared only in the middle- and high-elevation bands, and always in the upper range of “middle” elevations (1,200 m and above). There is high turnover in the frog fauna between the low- and high-elevation bands on Gunung Katopasa, with only Polypedates iskandari ranging both below 700 m and above 1,400 m. Limnonectes sp. “I” exhibits a similar distribution, and we encountered this frog up to 1,339 m in elevation, suggesting that it may occur above 1,400 m elsewhere on Gunung Katopasa. The same is likely true of Limnonectes sp. “T yellow”, which we recovered at 1,364 m. Sphenomorphus zimmeri inhabits only the high-elevation environments on Gunung Katopasa, whereas other Sphenomorphus species (S. variegatus and S. tropidonotus) are absent from higher elevation sites. These patterns of turnover in frogs and in Sphenomorphus are similar to those seen on Gunung Galang (Karin et al., 2023).

Given the 26–28 species estimated to be present on Gunung Katopasa but absent from our surveys, it seems possible that undescribed herpetofaunal species remain to be discovered on the mountain. From our experience conducting fieldwork on the Eastern Peninsula of Sulawesi, we can account for at least 15 species we did not collect that are either recorded or likely to be present in the area of Gunung Katopasa. These include two frog species: Fejervarya cancrivora and Limnonectes sp. “J” (Setiadi et al., 2011); the lizards Bronchocoela celebensis, Cyrtodactylus batik, Gekko iskandari, Gekko monarchus, Gekko smithii, Hemidactylus platyurus, and Emoia atrocostata, as well as the snakes Calamaria boesemani, Rabdion forsteni, Coelognathus erythrurus, Cerberus schneideri, Malayopython reticulatus, and Indotyphlops braminus. As is typical, the total snake richness of the mountain is unclear based on our survey (Table 2).

While deforestation for cropland has removed a large amount of easily accessible habitat on Gunung Katopasa, the mountain and its watersheds remain primarily forested. Though their steep terrain likely limits the extent of human impact on Katopasa and its neighboring peak Buyu Kandela, we are not aware of any formal legal protections for these ecosystems. The closest protected area, Cagar Alam Morowali to the south, does contain several peaks above 2,000 m asl, including Gunung Tokala and Gunung Tambusisi, which may harbor similar herpetological assemblages. However, the high-elevation ecosystems in these mountains have likely never been contiguous, meaning that the Katopasa and Kandela upland ecosystems are likely highly endemic and may warrant legal protection. The vast majority of herpetofauna on the mountain do not appear to be widely exploited by locals (Akhsa, Ramadhanil & Anam, 2015). We believe that habitat loss from uncontrolled fire is likely to be the largest immediate threat to the ecosystems of Gunung Katopasa. If deforestation expands into upland habitats, it would certainly pose a significant threat to these potentially unique habitats. As it stands, deforestation in lowland habitat poses a huge threat to the extremely species-rich lower elevational band.

Conclusions

The mountains of Sulawesi are home to a diverse, endemic, and highly under-studied biota, and a full understanding of this biodiversity depends on systematic and targeted surveys, mountain-by-mountain. The recent description of several herpetofaunal species endemic to the Eastern Peninsula, (Brown, Supriatna & Ota, 2000; Iskandar, Arifin & Rachmansah, 2011; Iskandar, Rachmansah & Arifin, 2011; Hamidy et al., 2025), in concert with at least six undescribed species reported here, indicate that this is a region of high potential for species discovery. This characterization of Gunung Katopasa’s herpetofauna likely describes only a tiny fraction of the total biodiversity sampled by our expedition, and we hope that it can provide a baseline for future work on the mountain and for future study of the unique ecosystems of Sulawesi’s Eastern peninsula.

Supplemental Information

Supplemental Information 1 Photographs of selected species and localities.

Supplemental Information 2 Metadata for all herpetofauna encountered in this survey.

Supplemental Information 3 Abstract in Bahasa Indonesian.

We are grateful for the critical assistance and support of our larger field expedition team, which made this project a success. In particular, we thank Anang Achmadi, Mohammad Irham, and Pungki Lupiyaningdya for coordinating the expedition, and Dede Avandi for his help managing permits and visas. We thank our field coordinator, Boy, for working to establish the field camps. We further thank the people of the village of Mire for their support and assistance in making this expedition possible, including their help in carrying thousands of pounds of gear and equipment up the mountain and their assistance making exceptional field camps. We also thank Carol Spencer and her curatorial team for their help importing specimens and depositing them in the MVZ collection.

Additional Information and Declarations

Competing Interests

The authors declare that they have no competing interests.

Author Contributions

Isaac W. Krone conceived and designed the experiments, performed the experiments, analyzed the data, prepared figures and/or tables, authored or reviewed drafts of the article, and approved the final draft.

Benjamin R. Karin conceived and designed the experiments, performed the experiments, analyzed the data, prepared figures and/or tables, authored or reviewed drafts of the article, and approved the final draft.

Jeffrey H. Frederick conceived and designed the experiments, performed the experiments, analyzed the data, prepared figures and/or tables, authored or reviewed drafts of the article, and approved the final draft.

Sina S. Amini analyzed the data, authored or reviewed drafts of the article, data Curation, and approved the final draft.

Simon G. Scarpetta performed the experiments, authored or reviewed drafts of the article, and approved the final draft.

Amir Hamidy performed the experiments, authored or reviewed drafts of the article, and approved the final draft.

Syahfitri Anita performed the experiments, authored or reviewed drafts of the article, and approved the final draft.

Awal Riyanto performed the experiments, authored or reviewed drafts of the article, and approved the final draft.

Evy Arida performed the experiments, authored or reviewed drafts of the article, and approved the final draft.

Wahyu Tri Laksono performed the experiments, authored or reviewed drafts of the article, and approved the final draft.

Umilaela Arifin analyzed the data, prepared figures and/or tables, authored or reviewed drafts of the article, and approved the final draft.

Bryan H. Bach analyzed the data, authored or reviewed drafts of the article, and approved the final draft.

Collin Bos analyzed the data, authored or reviewed drafts of the article, data Curation, and approved the final draft.

Charlotte K. Jennings conceived and designed the experiments, analyzed the data, authored or reviewed drafts of the article, data Curation, and approved the final draft.

Alexander L. Stubbs performed the experiments, authored or reviewed drafts of the article, and approved the final draft.

Kinsington Peterson analyzed the data, authored or reviewed drafts of the article, data Curation, and approved the final draft.

Sucheng Shi analyzed the data, authored or reviewed drafts of the article, data Curation, and approved the final draft.

Jimmy A. McGuire conceived and designed the experiments, performed the experiments, prepared figures and/or tables, authored or reviewed drafts of the article, data Curation, and approved the final draft.

Ethics

The following information was supplied relating to ethical approvals (i.e., approving body and any reference numbers):

The animal study protocol was approved by the Institutional Animal Care and Use Committee of the University of California Berkeley.

Field Study Permissions

The following information was supplied relating to field study approvals (i.e., approving body and any reference numbers):

Research and collection permits were granted by LIPI and RISTEKDIKTI (now BRIN).

Data Availability

The following information was supplied regarding data availability:

The figures and code are available in the Zenodo: dibamus. (2025). dibamus/Katopasa: Zenodo release (1.0.1). Zenodo. https://doi.org/10.5281/zenodo.15400761.

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
