# Peer review of "Elevational surveys of Sulawesi herpetofauna 2: Mount Katopasa on the Eastern Peninsula of Sulawesi island, Indonesia"

_PeerJ, doi:10.7717/peerj.20024_

## Round 0.1 · original submission · Major Revisions

Thank you very much for your manuscript titled “Elevational surveys of Sulawesi herpetofauna 2: Gunung Katopasa on the Eastern Peninsula of Sulawesi island, Indonesia” that you sent to PeerJ.

This study presents very valuable and relevant information on the herpetofauna present on an island in Indonesia.

As you will see below, comments from referee 1 suggest a minor revision while reviewers 2 suggests a major revision. Given this, I would like to see a major revision dealing with the comments. Their comments should provide a clear idea for you to review, hopefully improving the clarity and rigor of the presentation of your work. I will be happy to accept your article pending further revisions, detailed by the referees.

Reviewer 1 has some comments on the background, discussion and conclusion.

Reviewer 2's observations are noted in each of the sections of the manuscript.

Please note that we consider these revisions to be important and your revised manuscript will likely need to be revised again.

Reviewer 1 ·

Basic reporting

Sentence structure and punctuation need to be checked

Experimental design

-

Validity of the findings

-

Additional comments

The author needs to add the implications of this research in the discussion section.

Annotated reviews are not available for download in order to protect the identity of reviewers who chose to remain anonymous.

Reviewer 2 ·

Basic reporting

Why is herpetofauna divided into 3 classes namely frogs, lizards and snakes? While it is thoroughly divided into 2 classes namely amphibia and reptiles. Why not divide into frogs, toads, caesila for amphibia and skink, gecko, lizard, agamid lizard, and snake for reptiles?

Introduction:
1. What makes the biota of Sulawesi unique is its location in the central area divided by the Wallace line dividing the western zone and the Weber line dividing the western zone, where the Wallace and Weber lines are imaginary lines that divide the diversity in Indonesia. In the introduction only the Wallace Line is explained.
2. Review the appropriateness, relevance, and connection of the line to the title and theme of the article regarding 'elevational surveys of herpetofauna'.

Experimental design

Materials and Method:
1. The time of day to conduct the survey is not listed, whether it is morning, afternoon, evening, night, midnight, early morning or all day, as herpetofauna are diurnal and nocturnal.
2. How many pitfall buckets and how long is the drift-fence installed? Not described here. If there is documentary evidence, it can be included in the appendix.
3. How many glue trap lines were installed? Not explained here. If there is documentary evidence, it can be included in the appendix.
4. Complete the sentence in the section “and can be downloaded at ()”
5. Why are pitfall bucket, drift-fence, glue trap lines methods chosen? While there are still methods such as quadrat / micro habitat plots for more specific searches for herpetofauna.

Validity of the findings

Result and Discussion:
1. Installation of 30 sticky traps at post 6 and post 7 is written in the method not in the results section or can write the findings of the species of 30 sticky traps only in the results section.
2. What is the relation of “Forest loss” in the Result with the title and theme of “elevational survey of herpetofauna” so that it needs to be included in the Result chapter?
3. Making a location map must be adjusted to the rules of cartography. The map attached is incomplete with title, cardinal directions and others.
4. The line in the graph is not given a legend so it is not clear what it shows.
5. Ingerophrynus biporcatus is not present in the study site when looking at the IUCN distribution map. This species is found in lowland locations and is more abundant in the western part of Sulawesi island.
6. Was Ophiophagus hannah found in person? This is because there are many errors in identifying species if only based on interviews.

Additional comments

No comment

Annotated reviews are not available for download in order to protect the identity of reviewers who chose to remain anonymous.

---

## Round 0.2 · accepted · Accept

After reviewing this revised version of your manuscript, I see that the main comments suggested by the reviewers have been included, while the suggestions not considered are justified in detail. Therefore, I am satisfied with the current version and consider it ready for publication.

Reviewer 1 ·

Basic reporting

The manuscript has been revised according to suggestions so that it is clear.

Experimental design

The manuscript has been revised according to suggestions so that it is clear.

Validity of the findings

The manuscript has been revised according to suggestions so that it is clear.

Additional comments

no comment